# A Narrative Review of Preclinical In Vitro Studies Investigating microRNAs in Myocarditis

Olga Grodzka [1,2,†], Grzegorz Procyk [2,3,†] and Małgorzata Wrzosek [4,5,*]

1 Department of Neurology, Faculty of Medicine and Dentistry, Medical University of Warsaw, 80 Ceglowska St., 01-809 Warsaw, Poland; olga.grodzka@wum.edu.pl
2 Doctoral School, Medical University of Warsaw, 81 Żwirki i Wigury St., 02-091 Warsaw, Poland; grzegorz.procyk@wum.edu.pl
3 1st Chair and Department of Cardiology, Medical University of Warsaw, 1A Banacha St., 02-097 Warsaw, Poland
4 Department of Biochemistry and Pharmacogenomics, Medical University of Warsaw, 1 Banacha St., 02-097 Warsaw, Poland
5 Centre for Preclinical Research, Medical University of Warsaw, 1B Banacha St., 02-097 Warsaw, Poland
* Correspondence: malgorzata.wrzosek@wum.edu.pl
† These authors contributed equally to this work and are both first co-authors (O.G. and G.P.).

**Abstract:** According to the World Health Organization's statement, myocarditis is an inflammatory myocardium disease. Although an endometrial biopsy remains the diagnostic gold standard, it is an invasive procedure, and thus, cardiac magnetic resonance imaging has become more widely used and is called a non-invasive diagnostic gold standard. Myocarditis treatment is challenging, with primarily symptomatic therapies. An increasing number of studies are searching for novel diagnostic biomarkers and potential therapeutic targets. Microribonucleic acids (miRNAs) are small, non-coding RNA molecules that decrease gene expression by inhibiting the translation or promoting the degradation of complementary mRNAs. Their role in different fields of medicine has been recently extensively studied. This review discusses all relevant preclinical in vitro studies regarding microRNAs in myocarditis. We searched the PubMed database, and after excluding unsuitable studies and clinical and preclinical in vivo trials, we included and discussed 22 preclinical in vitro studies in this narrative review. Several microRNAs presented altered levels in myocarditis patients in comparison to healthy controls. Moreover, microRNAs influenced inflammation, cell apoptosis, and viral replication. Finally, microRNAs were also found to determine the level of myocardial damage. Further studies may show the vital role of microRNAs as novel therapeutic agents or diagnostic/prognostic biomarkers in myocarditis management.

**Keywords:** in vitro studies; inflammation; microRNAs; myocarditis; preclinical studies

## 1. Introduction

### 1.1. Myocarditis—Basic Information

The World Health Organization statement defines myocarditis as an inflammatory disease of the myocardium, diagnosed using histological, immunological, or immunohistochemical criteria [1]. According to the European Society of Cardiology's statement, endomyocardial biopsy remains the gold standard for a definite diagnosis [2]; however, cardiac magnetic resonance imaging is increasingly recommended as the preferable non-invasive approach [3]. Various etiologies are listed among causes that may lead to developing myocarditis, headed by infections, mainly viral, and autoimmune conditions [4]. Notably, the most commonly affected group, in contrast to other well-known cardiac diseases, such as myocardial infarction or heart failure, are young adults, with a slight dominance of men [5]. Noteworthily, myocarditis was indicated as one of the leading causes of sudden cardiac death and dilated cardiomyopathy [6]. However, it is hard to estimate the prevalence

worldwide due to the diverse clinical presentation. According to the reports, it varies between 0.12% and 12% [5].

Although much progress has been made in recent years in myocarditis diagnosis [7,8] with the development of novel non-invasive diagnostic tools [7], there remains a lack of treatment-focused studies [9]. Therefore, therapy remains mainly supportive and symptomatic, focusing on hemodynamic stabilization [10]. The long-term prognosis depends primarily on the cause of the disease [3,9]. Establishing a targeted therapy would allow us to provide patients with optimal management and lead to better long-term outcomes.

### 1.2. microRNAs—Small Regulators of Gene Expression

Microribonucleic acids (microRNAs, miRNAs, and miRs) are small, non-coding RNA molecules that regulate gene expression [11]. They consist of about 20–25 nucleotides [12] and fulfil their molecular role through binding to the complementary sequences of messenger RNAs (mRNAs), thus promoting mRNA degradation or inhibiting translation. In turn, protein synthesis is blocked [13]. MicroRNAs can be found and measured in different body fluids, such as blood, urine, or cerebrospinal fluid.

Briefly, miRNA biosynthesis starts in the nucleus and continues in the cytoplasm, where Drosha and Dicer, two RNase III proteins, process premature molecules [14,15]. In the final step, the guide strand remains and represents mature miRNA, while a passenger strand is discarded [16]. This is the canonical way and results in the formation of the 5′ miRNA involved in regulating gene expression. Most miRNAs are produced likewise; nevertheless, the different pathways independent of Drosha and Dicer are also known [17]. In turn, the 3′ miRNA can also be formed to bind with complementary mRNAs. This review includes information about the miRNA's end whenever the authors clearly state it.

MiRNAs have been thoroughly studied in recent years. Their putative role in multiple diseases in diverse fields of medicine was described, including cardiology [18,19], oncology [20,21], neurology [22], and autoimmune diseases [23]. Researchers often investigated the involvement of miRNAs in the pathogenesis of the disease, trying to find cause-and-effect relationships. Nevertheless, in some cases, they solely explored miRNAs as potential disease biomarkers without searching for an underlying cause of uncovered miRNA deregulation.

### 1.3. Suggested Value of MicroRNAs in Myocarditis

MiRNAs are altered in several cardiological conditions, including myocarditis. A recent study showed that not only circulating but also exosome-derived miRNAs may play a vital role in the mechanisms of myocarditis [24]. Regarding usually uncertain diagnoses [22] and mainly symptomatic treatment without targeted therapy options [25], microRNAs are considered a chance to facilitate myocarditis management. Several clinical and preclinical studies analyzed miRNAs' potential role in myocarditis, drawing promising conclusions. We aimed to discuss all of these studies comprehensively.

In previous reviews, we have already summarized clinical trials and in vivo preclinical trials [26,27]. In this review, the last one from the series about microRNAs in myocarditis, we aimed to analyze preclinical in vitro studies thoroughly. Apart from the diagnostic utilities of microRNAs, we also aimed to focus on the possibilities that miRNAs bring to myocarditis therapy.

## 2. Methodology

The role of miRNAs in myocarditis has been investigated in multiple studies. Therefore, only preclinical in vitro trials have been included in this narrative review to narrow the list of suitable research items and allow for a more accurate and solid discussion. Commentaries, letters to the editors, case reports, reviews, and clinical and preclinical in vivo studies were excluded. Articles written in languages other than English were not included.

We searched the PubMed database with the following query: "(miRNA OR microRNA) AND (myocarditis)". After removing inappropriate studies by title, type, or abstract, we

were left with 73 articles regarding the role of miRNAs in myocarditis. Excluding all of the papers other than those of original preclinical in vitro studies yielded 22 papers to be included in this review (Figure 1).

**Identification of studies via PubMed database**

Records identified from:
• PubMed Database

Search strategy: "(microRNA OR miRNA) AND (myocarditis)"

Records screened → Records excluded by title or type

Reports assessed by abstract ($n$=73) → Clinical ($n$=21) and preclinical in vivo ($n$=30) studies excluded

Studies included in the review ($n$=22)

Identification | Screening | Included

**Figure 1.** The flowchart for the selection process. miRNA—microRNA, $n$—the number of studies, and RNA—ribonucleic acid.

All included studies were divided into five paragraphs: (i) microRNA alterations in myocarditis, (ii) microRNA influence on myocardial inflammation, (iii) viral replication dependence on microRNAs, (iv) microRNA impact on cell apoptosis, and (v) in vitro models presenting the role of microRNAs in cardiac dysfunction.

## 3. MicroRNAs in an In Vitro Model of Myocarditis

### 3.1. microRNA Alterations in Myocarditis

Some in vitro studies presented clearly that miRNAs are altered significantly in the myocarditis model. Yao et al. [28] conducted a study investigating HeLa cells infected with CVB3 (Coxsackievirus B3) and compared them to non-infected cells. It was shown that the miR-107 level was elevated in the study group in comparison to the control group. The higher expression of miR-107 was related to CVB3 infection and associated with viral replication. Liu et al. [29] carried out a study consisting of an in vivo and in vitro part. In this review, we focused on the latter. It was shown that CVB3-infected HeLa cells demonstrated higher levels of miR-324-3p than non-infected cells. Lin et al. [30] studied CVB3-infected HL-1 cells. It was demonstrated that these cells presented increased levels of miR-19b compared to mock-infected HL-1 cells. All studies mentioned in this paragraph, with some additional information, are summarized in Table 1.

**Table 1.** A summary of studies focusing on microRNA alterations in myocarditis.

| Ref. | Year | Population | Comparison | miRNA | Outcome | Methodology |
|---|---|---|---|---|---|---|
| Yao et al. [28] | 2020 | CVB3-infected HeLa cells | normal HeLa cells | miR-107 | ↑ miR-107 in CVB3-infected HeLa cells | miRs by PCR |
| Liu et al. [29] | 2022 | CVB3-infected HeLa cells | normal HeLa cells | miR-324-3p | ↑ miR-324-3p in CVB3-infected HeLa cells | miRs by qPCR |
| Lin et al. [30] | 2016 | CVB3-infected HL-1 cells | mock-infected HL-1 cells | miR-19b | ↑ miR-19b in CVB3-infected HL-1 cells | miRs by microarray analysis and qPCR |

↑—increased, CVB3—Coxsackievirus B3, miR—microRNA, qPCR—quantitative polymerase chain reaction, ref.—reference, and RNA—ribonucleic acid.

### 3.2. microRNA Influence on Myocardial Inflammation

Several studies focused on miRNA impact on cardiac inflammation, the pathogenic base of myocarditis. Pan et al. [31] analyzed human cardiomyocytes that had been exposed to lipopolysaccharide (LPS) to induce myocarditis. Researchers measured the levels of interleukin 1β (IL-1β) and interleukin 6 (IL-6) to assess pyroptosis and cardiac inflammation. The levels of inflammatory factors were reduced in cells treated with miR-223-3p encapsulated in extracellular vesicles compared to the control group without any intervention. Zhu et al. [32], similar to the previous researchers, studied LPS-induced cardiomyocytes in which *SOX2* overlapping transcript (SOX2OT), which substantially contributes to heart damage, was silenced. An miR-215-5p inhibitor was transfected into the cells to abolish the suppressive role of SOX2OT-knockdown on IL-6 and tumor necrosis factor α (TNF-α) production. This intervention restored the elevated levels of these inflammatory factors in cells primarily caused by induction with LPS. After all, miR-215-5p was shown to play an anti-inflammatory role. Another study in which human cardiomyocytes were treated with LPS to imitate viral myocarditis was performed by Wang et al. [33]. Due to this intervention, the level of miR-16 was profoundly downregulated. These cells were then transfected with miR-16 mimic and compared to (i) normal cells, (ii) LPS-induced cells, and (iii) LPS-induced cells treated with an miR-16 inhibitor. Cardiomyocytes treated with LPS presented considerably higher levels of inflammatory factors—IL-6, interleukin 8 (IL-8), and TNF-α. However, it was repressed by miR-16 mimic administration.

Fei et al. [34] showed that the miR-146a level was elevated in CVB3-infected HeLa cells in comparison to non-infected ones. Moreover, it was proved that adding an miR-146a mimic exerted an anti-inflammatory effect on CVB3-infected cells. The group treated with this miR mimic showed decreased levels of inflammatory cytokines (IL-6 and TNF-α) compared to cells treated with the miR-146a antagonist. Toll-like receptor 3 (detecting viral RNA) was identified as a main target for miR-146a. Fan et al. [35] also conducted research using the HeLa cells model. It was shown that CVB3-infected cells presented elevated levels of miR-181d and miR-30a compared to control cells. Treating infected cells with miR-181d or miR-30a mimics caused an increase in the level of IL-6, while transfecting cells with inhibitors of these miRNAs caused a decrease in IL-6 levels. Chen et al. [36] compared the levels of inflammatory cytokines in CVB3-infected HeLa cells treated with either a mimic or an antagonist of miR-214. Transfection with miR-214 mimic was shown to increase the levels of TNF-α and IL-6. MiR-214 inhibition might be a possible strategy in myocarditis treatment.

A slightly different study was conducted by Chen et al. [37]. Apart from the in vivo part of their study, the researchers also isolated B-cells from the hearts of mice with myocarditis. Wild B-cells with normal miR-98 expression were compared to miR-98-deficient ones. LPS alone and LPS with TNF-α were added to both cultures. It was demonstrated that in LPS-treated cultures, the level of IL-10 was increased. The presence of TNF-α abolished this effect but only in wild B-cells. In B-cells presenting reduced levels of miR-98, the presence of TNF-α did not impact the IL-10 increase caused by LPS. Overall, miR-98 was shown to play a protective role in myocarditis. As concluded by the authors, miR-98 might be considered a potential therapeutic target in the treatment of myocarditis. All

studies mentioned in this paragraph, with some additional information, are summarized in Table 2.

**Table 2.** A summary of studies regarding the influence of microRNAs on inflammation.

| Ref. | Year | Population | Comparison | miRNA | Outcome | Methodology |
|---|---|---|---|---|---|---|
| Pan et al. [31] | 2022 | LPS-induced HCM treated with EVs with miR-223p | LPS-induced HCM (without intervention) | miR-223-3p | Cardiac inflammation and pyroptosis restriction caused by miR-223-3p. | IL-1β and IL-6 by ELISA |
| Zhu et al. [32] | 2022 | SOX2OT-silenced LPS-induced HCM + miR-215-5p inhibitor | SOX2OT-silenced LPS-induced HCM | miR-215-5p | ↑ IL-6 and TNF-α (IFs) in HCM with miR-215-5p inhibitor. | IFs by WB |
| Wang et al. [33] | 2020 | miR-16 mimic-treated LPS-induced HCM | normal HCM, LPS-induced HCM, or miR-16 inhibitor-treated LPS-induced-HCM | miR-16 | ↓ miR-16 in LPS cells compared to others; ↓ IL-6, IL-8, and TNF-α (IFs) in LPS-cells treated with miR-16 compared to others. | miRs by qPCR IFs by ELISA |
| Fei et al. [34] | 2020 | miR-146a mimic-treated CVB3-infected HeLa cells | normal HeLa cells or miR-146a antagonist-treated CVB3-infected HeLa cells | miR-146a | ↑ miR-146a in infected cells compared to others; ↓ IL-6 and TNF-α (Ifs) in cells treated with miR-146a mimic. | miRs by qPCR; Ifs by ELISA |
| Fan et al. [35] | 2019 | miR-181d or miR-30a inhibitors or mimic-treated CVB3-infected HeLa cells | normal HeLa cells, CVB3-infected HeLa cells, or miR-NC CVB3-infected HeLa cells | miR-181d miR-30a | ↑ miR-181d and miR-30a in infected HeLa cells compared to normal cells; ↑ IL-6 after injecting miR mimic; ↓ IL-6 after injecting miR inhibitor. | miRs by qPCR; IL-6 by ELISA |
| Chen et al. [36] | 2015 | miR-214 mimic-treated CVB3-infected HeLa cells | miR214 antagonist-treated CVB3-infected HeLa cells | miR-214 | ↑ TNF-α and IL-6 (IFs) in miR-214 mimic-treated HeLa cells. | IFs by ELISA |
| Chen et al. [37] | 2017 | LPS and LPS/TNF-α-treated miR-98-deficient B-cells | LPS and LPS/TNF-α-treated wild B-cells | miR-98 | ↑ IL-10 in LPS-treated wild B-cells (but not in LPS/TNF-α-treated wild B-cells); ↑ IL-10 in LPS and LPS/TNF-α-treated miR-98-deficient B-cells. | IL-10 by qPCR |

↑—increased, ↓—decreased, CVB3—Coxsackievirus B3, ELISA—enzyme-linked immunosorbent assay, HCM—human cardiomyocytes, Ifs—inflammatory factors, IL—interleukin, LPS—lipopolysaccharide, miR—microRNA, NC—negative control, qPCR—quantitative polymerase chain reaction, SOX2OT—SOX2 overlapping transcript, ref.—reference, RNA—ribonucleic acid, TNF—tumor necrosis factor, and WB—Western blotting.

### 3.3. Viral Replication Dependence on microRNAs

Germano et al. [38] studied the role of miR-590-5p in promoting viral infection. It was presented that cardiomyocytes exposed to CVB had significantly higher levels of miR-590-5p than normal cells. However, the injection of antagomiR-590-5p was shown to decrease viral load and replication. Another microRNA promoting viral replication was explored by Ye et al. [39], who also conducted a study on CVB3-infected HeLa cells. To analyze the role of miR-126 in viral myocarditis, the cells were treated with a mimic, an inhibitor, or a negative control of miR-126. Administration of an miR-126 mimic caused an increase in viral replication. Accordingly, the injection of an miR-126 inhibitor decreased the viral load. Overall, it was shown that miR-126 promoted CVB3 myocarditis.

Similarly, He et al. [40] conducted a study with HeLa cells infected with CVB3. It appeared that the viral replication was suppressed after miR-21 injection as compared to non-mammal miR. Liu et al. [29] modified CVB3-infected HeLa cells to overexpress miR-343-3p or to knockdown this miR expression. Cells overexpressing miR-343-3p had lower levels of viral capsid protein 1, a marker of viral replication, than miRNA-silenced ones. He et al. [41] engineered CVB3 to contain target sequences for miRNAs important in suppressing viral myocarditis (miR-133 and miR-206). Afterward, these modified viruses were injected into HeLa cells. As a result, viral replication was lowered, and cell viability was increased compared to the controls treated with either CVB3 containing negative control miRNA target sequences or non-modified CVB3. Considering the above, CVB3 containing target sequences for miR-133 or miR-206 is a promising candidate for CVB3 vaccines. Finally, Corsten et al. [42] investigated neonatal rat cardiomyocytes infected with

CVB3. The cells were then treated with an inhibitor, a mimic, and a scrambled control of the miR-221/-222 cluster. After inhibitor addition, the viral replication was elevated. Consistently, transfection with the mimic of the miR-221/-222 cluster caused a decrease in viral load. All studies mentioned in this paragraph, with some additional information, are summarized in Table 3.

**Table 3.** A summary of studies showing the effect of administrating microRNA mimics or antagonists on viral replication.

| Ref. | Year | Population | Comparison | miRNA | Outcome | Methodology |
|---|---|---|---|---|---|---|
| Germano et al. [38] | 2019 | antagomiR-590-5p-treated CVB-infected CM | normal CM miR-590-5p mimic-treated CVB-infected CM. | miR-590-5p | ↑ miR-590-5p in infected CM compared to normal CM; ↓ viral load and ↓ VP1 in antagomiR-590-5p-treated CVB-infected CM compared to other infected cells. | miRs by qPCR; viral load by plaque assay; VP1 by WB. |
| Ye et al. [39] | 2013 | miR-126 mimic-treated CVB3-infected HeLa cells | miR-126 inhibitor-treated CVB3-infected HeLa cells; miR-NC-treated CVB3-infected HeLa cells. | miR-126 | ↑ miR-126 after CVB3 infection VP1 and viral replication; ↑ miR-126 mimic treated and ↓ miR-126 inhibitor-treated cells compared to NC. | miRs by qPCR; VP-1 by WB; replication by plaque assay. |
| He et al. [40] | 2019 | miR-21-treated CVB3-infected HeLa cells | non-mammal-miR-treated CVB3-infected HeLa cells. | miR-21 | ↓ viral load in miR-21-treated CVB3-infected cells compared to others. | Viral load by plaque assay. |
| Liu et al. [29] | 2022 | miR-324-3p-overexpressing CVB3-infected HeLa cells | miR-324-3p-silenced CVB3-infected HeLa cells. | miR-324-3p | ↓ VP1 in miR-324-3p-overexpressing cells. | VP1 by WB. |
| He et al. [41] | 2015 | miR-133 or -206-Ts-engineered CVB3-infected HeLa cells | non-mammal-miR-Ts-engineered CVB3-infected HeLa cells; CVB3-infected HeLa cells. | miR-133 miR-206 | ↓ viral replication and ↑ cell viability in miR-133 or -206-Ts-engineered CVB3-infected cells. | Viral replication by plaque assay; cell viability by MTS assay. |
| Corsten et al. [42] | 2015 | miR-221/-222-inhibited CVB3-infected neonatal rat CMs | miR-221/-222 mimic-treated CVB3-infected neonatal rat CMs; miR-NC-treated CVB3-infected neonatal rat CMs. | miR-221/-222 | ↑ viral load in miR-221/-222-inhibited CM compared to other VMC CM. | Viral load by qPCR. |

↑—increased, ↓—decreased, CM—cardiomyocytes, CVB3—Coxsackievirus B3, miR—microRNA, miR-Ts—miRNA target sequences, MTS—3-(4,5-dimethylthiazol-2-yl)-5-(3-carboxymethoxyphenyl)-2-(4-sulfophenyl)-2H-tetrazolium, NC—negative control, qPCR—quantitative polymerase chain reaction, ref.—reference, RNA—ribonucleic acid, VMC—viral myocarditis, VP1—viral capsid protein 1, and WB—Western blotting.

### 3.4. microRNA Impact on Cell Apoptosis

Several studies explored not only the influence on inflammatory response or viral load but also how microRNAs affect apoptosis. Zhang et al. [43] analyzed human cardiomyocytes overexpressing miR-8055 and compared them to the miR-8055-silenced cardiomyocytes. Cells with miR-8055 overexpression presented decreased levels of inflammatory factors, myocardial injury biomarkers, and apoptotic cell ratio. Similarly, Xiang et al. [44] showed that H9c2 cells exposed to LPS and subsequently treated with an miR-27a mimic had ameliorated cell viability; thus, the apoptotic cell ratio was decreased (control cells were injected with an miR-27a inhibitor). Overall, it was proved that miR-27a was protective in LPS-damaged H9c2 cells. Li et al. [45] explored the influence of an miR-203 inhibitor and mimic on H9c2 cells exposed to LPS. The downregulation of miR-203 promoted cell survival, lowered apoptotic rate, and decreased levels of inflammatory cytokines. On the other hand, miR-203 overexpression exacerbated cell apoptosis and contributed to an increased inflammatory response. As suggested by the authors, miR-203 plays its role by inhibiting the expression of a nuclear factor interleukine-3 (NFIL3), a survival mediator in the heart.

Tong et al. [46] studied the role of miR-15 in CVB3-infected H9c2 cells. The miR-15 inhibitor was added to these cells and compared to three other groups: (i) normal H9c2 cells,

(ii) CVB3-infected H9c2 cells, and (iii) CVB3-infected cells treated with the negative control miR. It was demonstrated that the group receiving the miR-15 inhibitor presented lower levels of inflammatory factors—IL-1β, IL-6, and interleukin 18 (IL-18). Moreover, miR-15 inhibition promoted the viability of CVB3-infected cells. Xia et al. [47] also conducted the in vitro part of their study on CVB3-infected H9c2 cells. They demonstrated that both miR-217 and miR-543 inhibitors caused a reduction in cell apoptosis and mitigation of inflammatory response (measured as IL-1β and IL-6 levels). Li et al. [48] presented that CVB3-infected H9c2 cells showed restrained expression levels of miR-16 compared to non-infected cells. Furthermore, the levels of inflammatory factors (IL-6 and TNF-α) and cell apoptosis decreased in the viral myocarditis cell model after transfection with an miR-16 mimic.

Zhang et al. [49] performed the preclinical part of their study on human cardiomyocytes. They showed that treating these cells with an miR-98 mimic caused a decrease in the expression of *FAS* and *FASL* genes. Furthermore, the apoptotic cell ratio was decreased in cells overexpressing miR-98. All studies discussed in this paragraph, with some additional information, are summarized in Table 4.

**Table 4.** A summary of studies presenting the influence of microRNAs on apoptosis.

| Ref. | Year | Population | Comparison | miRNA | Outcome | Methodology |
|------|------|-----------|------------|-------|---------|-------------|
| Zhang et al. [43] | 2021 | miR-8055-overexpressing LPS-treated HCM | miR-8055-silenced LPS-treated HCM. | miR-8055 | ↓ IL-1β, IL-6, and TNFα (IFs); ↓ cTnT, CKMB, BNP, and apoptotic rate in miR-8055-overexpressing cells. | IFs, cTnT, CKMB, and BNP by ELISA; apoptotic cells by FC. |
| Xiang et al. [44] | 2019 | miR-27 mimic-treated LPS-H9c2 cells | miR-27 inhibitor-treated LPS-H9c2 cells. | miR-27 | ↓ cell apoptosis in miR-27 mimic-treated cells. | Cell apoptosis by MTT assay and FC. |
| Li et al. [45] | 2019 | miR-203 inhibitor-treated LPS-H9c2 cells | miR-203 mimic-treated LPS-H9c2 cells; miR NC-treated LPS-cells. | miR-203 | ↓ IL-6, IL-8 (IFs), and cell apoptosis after miR-203 inhibition. | Cell apoptosis and IFs by ELISA. |
| Tong et al. [46] | 2020 | miR-15 inhibitor-treated CVB3-infected H9c2 cells | normal H9c2 cells; CVB3-infected H9c2 cells; miR NC-treated CVB3-infected cells. | miR-15 | ↑ miR-15 in CVB3-infected cells compared to normal cells; ↓ cell apoptosis and ↓ IL-1β, IL-6, and IL-18 (Ifs) in cells treated with miR-15-inhibitor compared to other CVB3-infected cells. | miRs by qPCR; cell apoptosis by FC; Ifs by ELISA. |
| Xia et al. [47] | 2020 | miR-217 inhibitor- or miR-543 inhibitor-treated CVB3-infected H9c2 cells | normal H9c2 cells; miR NC-treated CVB3-infected H9c2 cells. | miR-217 miR-543 | ↑ miR-217 and miR-543 in CVB3-infected H9c2 cells compared to normal cells; ↓ cell apoptosis, ↓ IL-1β, and IL-6 (Ifs) caused by miR inhibitor. | miRs by qPCR; cell apoptosis by FC; Ifs by ELISA. |
| Li et al. [48] | 2020 | miR-16 mimic-treated LPS-induced H9c2 cells | normal H9c2 cells; NC-treated LPS-induced-H9c2 cells. | miR-16 | ↓ miR-16 in LPS-induced cells than in normal cells; ↓ cell apoptosis, IL-6, and TNF-α (IFs) in LPS-induced cells treated with miR-16 compared to other LPS-induced cells. | miRs by qPCR; cell apoptosis by FC; IFs by WB. |
| Zhang et al. [49] | 2016 | miR-98 mimic-treated HCM | miR-98 inhibitor-treated HCM; miR NC-treated HCM. | miR-98 | ↓ *FAS* and *FASL* mRNA and ↓ apoptotic cells in HCM treated with miR-98 mimic compared to others. | miRs, *FAS*, and *FASL* by qPCR; apoptotic cells by FC. |

↑—increased, ↓—decreased, BNP—brain natriuretic peptide, CKMB—creatine phosphokinase MB, cTnT—cardiac troponin T, CVB3—Coxsackievirus B3, ELISA—enzyme-linked immunosorbent assay, FC—flow cytometry, HCM—human cardiomyocytes, Ifs—inflammatory factors, IL—interleukin, LPS—lipopolysaccharide, miR—microRNA, MTT—3-(4,5-dimethylthiazol-2-yl)-2,5-diphenyl tetrazolium bromide, NC—negative control, qPCR—quantitative polymerase chain reaction, ref.—reference, RNA—ribonucleic acid, TNF—tumor necrosis factor, and WB—Western blotting

### 3.5. In Vitro Models Presenting the Role of microRNAs in Cardiac Dysfunction

Xu et al. [50] investigated cardiomyocytes derived from neonatal rats, aiming to assess the influence of overexpressed miR-1 on the level of connexin 43 (Cx43)—an important transmembrane protein. It was observed that the upregulation of miR-1 decreased Cx43 expression. The decrease in Cx43 may lead to interference with cardiac function in my-

ocarditis. In the previously mentioned study [30], researchers transfected human-induced pluripotent stem cell-derived cardiomyocytes (hiPSCs-CMs) with the miR-19b mimic since it was shown to be upregulated in the VMC model. Irregular beats and decreased beating rates were observed as a result of this intervention. Gap junction protein α1, responsible for the electrical synchrony of cardiomyocytes, was indicated as a target for miR-19b. Thus, the dysregulation of miR-19b might explain arrhythmia occurrence in viral myocarditis patients. Both studies mentioned in this paragraph, with additional information, are summarized in Table 5.

**Table 5.** A summary of studies showing the role of microRNAs in cardiac dysfunction.

| Ref. | Year | Population | Comparison | miRNA | Outcome | Methodology |
|------|------|-----------|------------|-------|---------|-------------|
| Xu et al. [50] | 2012 | miR-1-treated neonatal rat CM | miR-1-inhibited neonatal rat CM; miR NC-treated neonatal rat CM. | miR-1 | ↓ Cx43 protein in miR-1-treated CM compared to other CM | Cx43 by WB |
| Lin et al. [30] | 2016 | miR-19b mimic-treated hi-PSCs-CMs | NC-treated hi-PSCs-CMs. | miR-19b | Irregular beating patterns and decreased beating rate of CM treated with miR-19b mimic | Beating assessed in timeslot |

↓—decreased, CM—cardiomyocytes, Cx—connexin, hi-PSCs-CMs—human-induced pluripotent stem cell-derived cardiomyocytes, miR—microRNA, NC—negative control, ref.—reference, and WB—Western blotting.

## 4. Conclusions and Future Perspectives

MiRNAs are being increasingly investigated in many diseases in various fields of medicine, thus creating novel opportunities for their better management. Amongst them, myocarditis has also been thoroughly researched. Here, we have presented different perspectives from preclinical in vitro trials on the value of miRNAs in myocarditis. MiRNAs influence mechanisms in myocarditis, such as inflammation, apoptosis, and viral replication. Moreover, particular miRNAs are observed to have different levels in the groups of patients and healthy controls. It is crucial not only for myocarditis diagnostic options but also for targeted therapies using either mimics or antagonists of miRNAs (Figure 2).

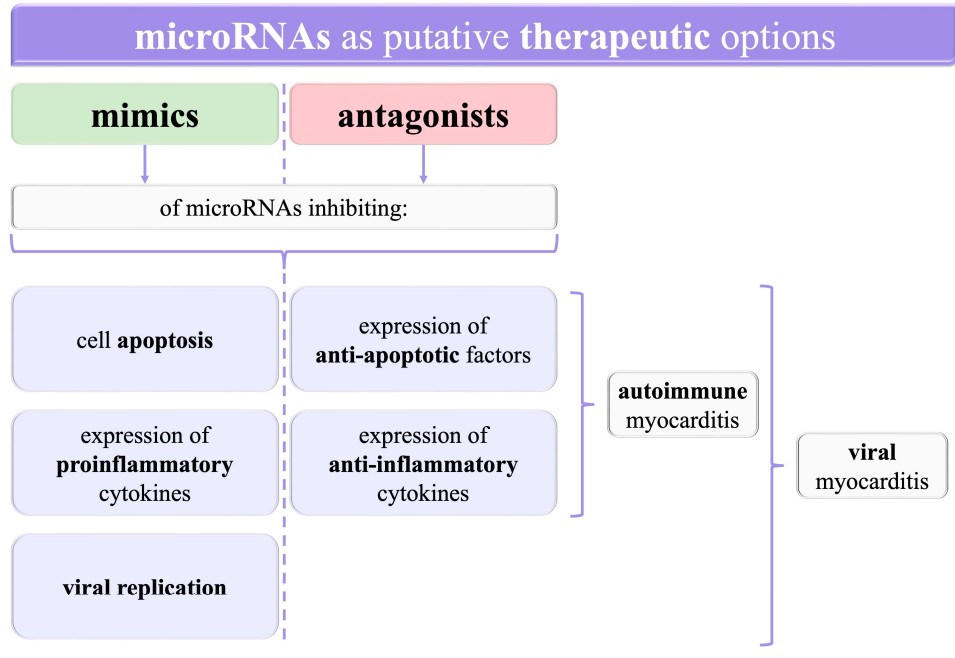

**Figure 2.** A graphical summary of the opportunities to use microRNAs as potential targets for therapeutic agents. RNA—ribonucleic acid.

Since microRNAs have been shown to influence the levels of diverse proteins, such as inflammatory factors, further investigations into myocarditis should bring new insights into the molecular bases of the disease. Since the treatment remains mainly symptomatic in both viral and autoimmune myocarditis, with attempts to administer antiviral drugs in the former and immunosuppressives in the latter, a better understanding of the pathogenesis may increase the role of targeted therapy. For instance, mimics of miRNAs decreasing the expression of proinflammatory cytokines, as well as antagonists of miRNAs exerting an anti-inflammatory effect, can presumably alleviate inflammation directly in the myocardium. This could enable the avoidance of harmful systemic adverse reactions of immunosuppressive drugs.

Some miRNAs may be crucial in terms of viral replication. Administrating inhibitors of those miRNAs that were shown to increase viral load might be a potential treatment for viral myocarditis. Similarly, inducing overexpression of microRNAs which are related to decreased viral replication is another putative therapy option. Finally, medications based on microRNAs may be used to reduce cell apoptosis and therefore halt the deterioration of the myocardium and prevent impaired heart function after recovery from myocarditis.

Few in vitro studies focused on the correlation between miRNAs and cardiac function; nevertheless, they cannot be missed. Myocarditis, in some cases, leads to permanent impairment of cardiac function, for instance, cardiomyopathies, especially dilated cardiomyopathy, or potentially life-threatening arrhythmias. Therefore, finding a valuable prognostic biomarker would help assess the patient's prognosis and identify those patients who need a more comprehensive approach. As discussed, arrhythmia occurrence in viral myocarditis patients may be partially explained by the dysregulation of miR-19b, and thus, it emerges as another potential therapeutic target.

In conclusion, miRNAs can also be useful as prognostic biomarkers or in targeted therapy, in addition to their role in myocarditis diagnosis. However, more studies exploring this issue are needed. Investigations of the effects of mimics or antagonists of particular microRNAs in myocarditis models may bring us closer to better management of myocarditis with more specific approaches.

### 5. Limitations

First, we searched only the PubMed database, which might have led to omitting some studies in the field. Nevertheless, it should not cause any statistical or reasoning bias since we performed solely a literature overview in a narrative style. The lack of statistical analysis of discussed data is another limitation. The narrative nature of our review can also explain this. Due to the same reasons, we evaded the risk of bias analysis.

**Author Contributions:** Conceptualization, O.G. and G.P.; methodology, O.G.; writing—original draft preparation, O.G.; writing—review and editing, G.P.; visualization, O.G.; supervision, M.W. All authors have read and agreed to the published version of the manuscript.

**Funding:** The research was funded by grant no. 12/TJ_2/2023 to O.G., and grant no. 85/TJ_2/2023 to G.P. Both grants are funded within "Talenty Jutra" by "Fundacja Empiria i Wiedza".

**Conflicts of Interest:** The authors declare no conflicts of interest.

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
