# Peer review of "A Narrative Review of Preclinical In Vitro Studies Investigating microRNAs in Myocarditis"

_cimb, doi:10.3390/cimb46020091_

Round 1

Reviewer 1 Report

Comments and Suggestions for Authors

In this review, all relevant preclinical in vitro studies (22 studies) regarding miRNAs in myocarditis and their role have been discussed.

The review identifies miRNAs for further studies as novel therapeutic agents or diagnostic/prognostic biomarkers in myocarditis management.

As this is a review – there is no new information in the publication. Only summary of already existing information on miRNA studies.

The conclusions are consistent with the evidence and arguments presented. I finde the division of the results in different topics (myocarditis, myocardial inflammation, viral replication, cell apoptosiscardiac dysfuction) very helpful and informative.

The references are appropriate and quite new.

Figures (1 and 2) shouldn’t be so big. Tables are nicely done.

Very good written and understandable publication. No issues to mention.

Great work!

Reviewer 2 Report

Comments and Suggestions for Authors

One recent study that identified 14 upregulated and 14 downregulated exosome-derived miRNAs could be included in the set of studies summarized in Table 1. A summary of studies focusing on microRNAs’ alterations in myocarditis.

Wu Q, Huang C, Chen R, Li D, Zhang G, Yu H, Li Y, Song B, Zhang N, Li B, Chu X. Transcriptional and functional analysis of plasma exosomal microRNAs in acute viral myocarditis. Genomics. 2024 Jan;116(1):110775. doi: 10.1016/j.ygeno.2023.110775. Epub 2023 Dec 30. PMID: 38163573.

Suggestions for minor changes:

For the table 5, 2nd entry, last item: "beat assessed in time slot" is quite uncommon usage in English, probably replace that with "heart rate" or "heart beating rate" as mentioned in the original paper would be better.

Line 41, replace "infectious" with "infections.

Line 58 "play they molecular role " should be revised

Line 60, specify what is "They " do you mean MicroRNA?

Line 68 move the 'also' to later part of the sentence, i.e."can also be ..."

Line 53, the word "allow" should be followed by "us".

Line 47, use period instead of comma as per the American convention i.e. "0.12%" instead of "0,12%"

Line 284, bringing new "insides" probably should be "insights" instead.

Comments on the Quality of English Language

The quality of English needs to be improved: some of the errors have been pointed out as shown above, but probably the paper should be further checked or edited by a native English speaker.

Reviewer 3 Report

Comments and Suggestions for Authors

The article offers an interesting overview of the role of miRNA in the evaluation of myocarditis in preclinical studies.

- is this a truly systematic review? was registered in the prospero database?

- if it was a scoping or systematic review, did you follow the prisma checklist

- apparently, you included all the preclinical studies. please describe better the inclusion and exclusion criteria

- why do you only perform research on PubMed? usually, at least 2 databases are adopted

- how did you and which kind of data you extracted from each study?

- did you pre-specificate some outcomes for inclusion/exclusion of studies?

- please explain better how did you represented results and if and why you performed a statistical synthesis or not

- did you perform some analysis of the risk of bias?

Comments on the Quality of English Language

minor editing

Round 2

Reviewer 3 Report

Comments and Suggestions for Authors

no more issues